# Microstructure and Mechanical Properties of Sn58Bi Components Prepared by Laser Beam Powder Bed Fusion

Chuan Yang, Ding Ding, Kaihua Sun, Mingyan Sun, Jie Chen, Yingying Wang *, Yonghao Zhang * and Bowen Zhan

Institute of Machinery Manufacturing Technology, China Academy of Engineering Physics, Mianyang 621900, China; yangchuan_6s@caep.cn (C.Y.); dingd2010d@163.com (D.D.); sundoom@126.com (K.S.); yanmingsun18@caep.cn (M.S.); jiechen_6s@caep.cn (J.C.); zbw163163163@163.com (B.Z.)
* Correspondence: yingying89@caep.cn (Y.W.); zyhao_6s@caep.cn (Y.Z.); Tel.: +86-816-2490543 (Y.W. & Y.Z.)

**Abstract:** As a low melting point alloy, Sn58Bi alloy plays a unique role in many fields. However, the brittleness of Sn58Bi alloy is a limitation that has to be addressed for wider applications. According to previous studies, third element addition is favorable for improved ductility, which is attributable to structure refinement. Therefore, laser beam powder bed fusion technology was adopted to prepare Sn58Bi alloy components. Additionally, the as-printed specimens presented more refined structures compared to the as-cast specimen, so they showed better plasticity. The Sn58Bi alloy showed excellent formability when specimens were prepared by LBPBF between 40–80 W. The density of specimens nearly remained stable above laser power higher than 40 W, and CT scanning could not detect internal defects, so the Sn58Bi alloy specimen was likely to be well fabricated at high laser power. Within the laser power range of 40–80 W, with the variation of laser power or scanning velocity, the laser energy density changed accordingly, and the mechanical strength of specimens was improved with the increase of laser power density. This strength change was probably related to the microstructure evolution and internal residual stress in the printing process.

**Keywords:** Sn58Bi alloy; laser beam powder bed fusion; microstructure; mechanical property





## 1. Introduction

The SnBi alloy and other low melting point alloys have aroused great interest among researchers for many years in view of their distinct applications, including solder alloys, melting model casting, tube bulging, rapid mold preparation and so on [1–4]. Eutectic Sn58Bi (wt%) alloy with a melting point of 139 °C is a typical low melting point alloy, and it is considered potentially promising as a lead-free solder alloy [5,6]. However, the brittleness of Sn58Bi alloy is a limitation that has to be addressed for wide applications [7]. Brittleness of the alloy is mostly caused by the intrinsic brittle Bi phase, and a coarsened microstructure also causes the reduction of mechanical properties [8].

To solve this issue, two methods to prepare SnBi alloy samples were adopted: the addition of alloying elements for the as-cast alloy and severe plastic deformation. As-cast eutectic Sn58Bi alloys containing additives have been investigated by many researchers. Shiqi Zhou et al. reported that the ultimate tensile stress and yield stress of Ti added Sn58Bi alloy increased with increasing Ti content, but elongation decreased at the same time [9]. Yang et al. confirmed that Ni could improve the ultimate tensile stress and yield stress of Sn58Bi alloy by restraining microstructure coarsening [8]. In fact, the third element, including Zn, Ag, Cu Sb and In, were also added to the Sn58Bi alloy by researchers, and in regard to the tensile strength and breaking elongation results, it is clear that the third element addition is favorable for an improvement in strength and/or ductility [10,11]. The reason for properties improvement could be explained as follows: the addition of a third element increases the liquidus temperature and makes the formation of intermetallic compounds or primary crystals of the added element more likely, and the pre-existing inclusions (IMC or primary crystal) serves as heterogeneous nucleation sites in the matrix

during solidification, which effectively refines the microstructures. Therefore, the refinement of the Sn58Bi alloy microstructure was the key factor in enhancing the mechanical performance of as-cast samples.

Severe plastic deformation is another method used to improve mechanical performance. A cast SnBi eutectic alloy was processed by Wang C.T. et al. using high-pressure torsion (HPT) at room temperature, and the HPT SnBi alloy exhibited superplastic behavior with elongations of up to >1000% [12]. Alkorta J. and Sevillano J.G. found that the equal channel angular pressing (ECPT) process could improve the plasticity of SnBi eutectic alloy [13]. It is well known that ultrafine grain produced by severe plastic deformation is responsible for the improvement of mechanical performance.

From the addition of alloying elements and severe plastic deformation methods, it could be inferred that microstructure refinement could evidently improve the mechanical properties of SnBi alloy. However, the addition of alloying elements may change the melting point of solder paste alloy, while severe plastic deformation methods, such as ECAP and HPT, are not suitable for large bulk materials. Therefore, a new approach to improve mechanical performance without changing the melting point of SnBi alloy is urgently needed. According to a previous investigation, even a small variation in the cooling rate will lead to significant changes in the microstructure and properties of SnBi alloy [14], and a rapid cooling process probably leads to the fine structure of SnBi alloy. As we know, extremely rapid solidification and cooling processes are the unique characteristics of some additive manufacturing (AM) technologies, for example, LBPBF (laser beam powder bed fusion) technology. Thus, additive manufacturing technology was first used to attempt to fabricate Sn58Bi specimens in order to obtain fine structure and better mechanical performance.

Additive manufacturing technology can be used to fabricate components with complex structures directly and rapidly [15]. LBPBF provides improvements in product quality, processing time and manufacturing reliability compared to binder-based laser sintering AM processes. In fact, stainless steel 316 L, AlSi10Mg, $Cu_2O$ and many other materials have been used to fabricate components using LBPBF technology, and the influence of parameters on the microstructure and properties was also investigated by researchers [16–19]. As far as low melting point alloys are concerned, few studies on additive manufacturing have been conducted. The fused coating process was adopted by Guangxi Zhao et al. to fabricate Sn63Pb37 components, which had higher tensile strength than that of the cast specimen [20]. Rong Wenjuan et al. used fused deposition to prepare SnBi alloy materials, but the microstructure of the bottom layer, center layer and the upper layer were obviously different [16]. Mask deposition and direct writing of metal ink are also adopted for low melting point alloy printing on electronic circuitry [21,22], but it is not suitable for the preparation of large bulk materials. Therefore, until now, few researchers have addressed the problem of the microstructure refinement and performance enhancement of SnBi alloy by additive manufacturing, especially by LBPBF.

Given the high manufacture, precise and unique solidification process of LBPBF technology, we proposed LBPBF technology to prepare eutectic Sn58Bi alloy specimens. The rapid cooling process of Sn58Bi alloy during LBPBF will contribute to microstructure refinement, thereby improving mechanical properties. In this study, the influence of printing parameters, including scanning velocity and laser power, on the mechanical properties and microstructures were investigated. The relationships between microstructure and mechanical properties of Sn58Bi alloy under different LBPBF parameters were also discussed accordingly.

## 2. Materials and Methods

High purity Sn58Bi alloy powders (chemical analysis is given in Table 1) were used to prepare the Sn58Bi alloys. The powder, which was produced by a plasma atomization process, was spherical with a particle size between 15 and 50 μm (Figure 1). Half of the particle volume had a powder size smaller than 36.85 μm. All specimens were produced

by EOS M290 LBPBF equipment. This LBPBF installation was equipped with a fiber laser, which can reach a maximum power of 200 W in continuous mode. The intensity distribution can be assumed to be Gaussian, and the spot diameter was 50 μm. The scanning velocity can be varied from 100 to 3000 mm/s; the maximum diameter of the useful building area was about 100 mm. The building chamber was first evacuated and then filled with an inert argon atmosphere in order to create an atmosphere with a low oxygen content during building.

**Table 1.** Chemical compositions (wt%) of the Sn58Bi powder used and as-cast Sn58Bi alloy.

| Condition | Sn | Bi | In | Pb | Ge | Sb | O | Ag |
|---|---|---|---|---|---|---|---|---|
| Power | Bal. | 57.86 | 0.011 | 0.007 | 0.006 | 0.005 | 0.005 | 0.002 |
| As-cast | Bal. | 57.83 | 0.015 | 0.005 | 0.005 | 0.005 | 0.005 | 0.003 |

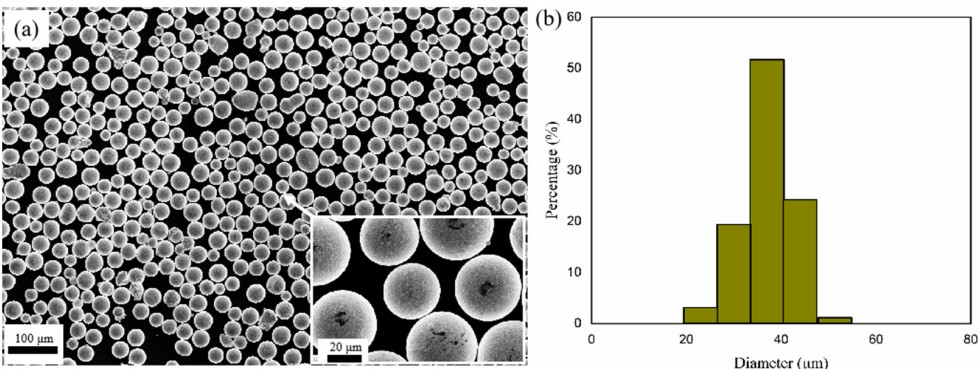

**Figure 1.** Sn58Bi alloy powders for LBPBF: (**a**) power morphology observed by SEM, (**b**) power diameters distribution.

In order to compare as-printed and as-cast specimens, the as-cast Sn58Bi alloy samples were also fabricated. Sn58Bi alloy pieces were melted and heated to 250 °C in a smelting furnace, and then the molten metal was poured into a sand mold at 190–200 °C. After cooling to room temperature and cutting the riser, the Sn58Bi alloy bars (ø30 × 150 mm) were obtained, which could be machined to the compression test specimens.

In this research, the influence of the scanning velocity $v$ and laser power $P$ on the properties of bulk material produced by LBPBF was investigated. Column specimens (ø6 × 10 mm) were produced by LBPBF. In view of the low melting point of Sn58Bi alloy, the scanning velocities in the experiment were set from 500 to 1500 mm/s, and laser powers were set from 30 to 80 W. The scanning strategy for the successive layers was set to rotate the scanning direction through 67° for the cross-hatching strategy. Layer thickness was set to 60 um, and hatch distance was set to 0.05 mm. The laser spot diameter was 0.06 mm. Preheating was not used before printing. Table 2 provide an overview of the parameters used for the various specimens. For specimens obtained, phase identification was carried out by standard X-ray diffraction (XRD) with Cu-Kα radiation. The microstructures were characterized using an optical microscope and scanning electron microscopes (SEM) equipped with an energy dispersive spectroscope (EDS). Computed tomography (CT) was employed to scan the defects within specimens using micro-CT. X-Ray-Tube voltage was 200 KV, X-Ray-Tube current was 0.11 mA, magnification was 11.31, 3D-Pixel size was 0.010 mm and the resolution was about 40 μm. After the surfaces of column specimens were polished using sand paper, the uniaxial compression test was conducted on an Instron 5985 test machine with a compressing speed of 0.5 mm/min under ambient temperature so as to investigate the influence of LBPBF parameters on the mechanical properties of different specimens.

**Table 2.** The overview of the process parameters for the different specimens.

| Specimen Group | A1 | A2 | A3 | B1 | B2 | B3 |
|---|---|---|---|---|---|---|
| Laser power (W) | 30 | 30 | 30 | 40 | 40 | 40 |
| Scanning velocity (mm/s) | 1500 | 1000 | 500 | 1500 | 1000 | 500 |
| Specimen group | C1 | C2 | C3 | D1 | D2 | D3 |
| Laser power (W) | 60 | 60 | 60 | 80 | 80 | 80 |
| Scanning velocity (mm/s) | 1500 | 1000 | 500 | 1500 | 1000 | 500 |

## 3. Results and Discussion

Figure 2a show the specimens prepared by LBPBF with parameters in Table 2. In fact, the experiment demonstrated that Sn58Bi alloy power could not be melted sufficiently and bonded into bulk material below laser power of 30 W at any scanning velocity (see Figure 2a), while the heavy dark smoke appeared above laser power of 80 W. Therefore, the laser powers were selected between 30 and 80 W. There were two specimens with the same parameter on each panel, and two panels of specimens were fabricated. In theory, the laser scanning velocity should be set slower when applied to higher laser power so as to apply the proper laser energy density on the power bed surface.

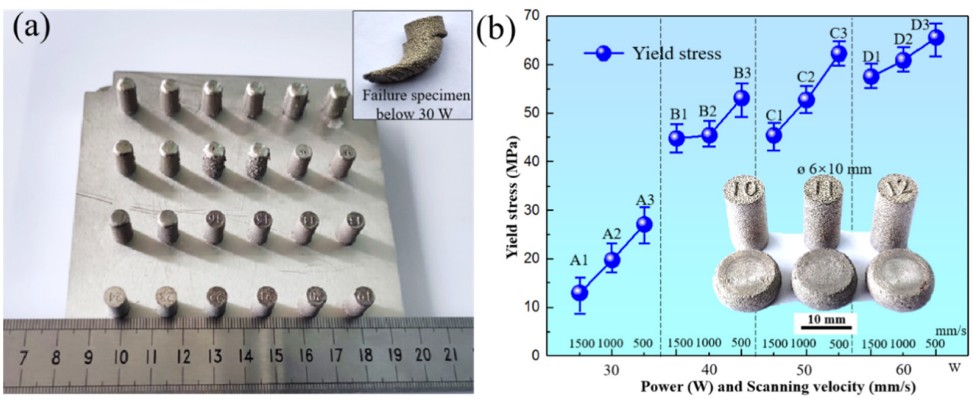

**Figure 2.** LBPBF Sn58Bi specimens (**a**) and uniaxial compression test results (**b**).

In the uniaxial compression test, the fracture of SnBi alloy specimens did not occur at all due to excellent plasticity, so the 80% reduction in height of the specimen was set to terminate the compression test. According to the result of the compression test (see Figure 2b), yield stress increased with the decrease of the scanning velocity under the same laser power. However, the yield stress of specimens printed with 30 W was much lower than those of specimens in other groups. Decrement of scanning velocity also increased the yield stress of specimens fabricated with the same laser power, which could be ascribed to an increment of energy density as same as laser power.

Since the porosity of LBPBF specimens is one of the important factors which affect the mechanical properties, the Archimedes method was used to measure the densities of Sn58Bi alloy specimens. Relative density variation with the change of printing parameters is illustrated in Figure 3b. The relative densities of as-printed specimens fabricated with 30 W laser power ranged from 85.0% to 93.8%, which were much smaller than other specimens. The optical pictures of the cross-section in Figure 3a showed that a mass of voids was easily observed on the surface, and the CT pictures also illustrated a mass of defects (dark shadow) that existed in the specimens, which indicated that specimens had high porosity under low laser power. In fact, the voids were caused by unfused power under low laser power (see Figure 3a), leading to low relative density. When the laser power was higher than 40 W, the relative densities of specimens dramatically increased to a value of around 98%. As one of the solder paste materials, the low melting point (138 °C) and excellent weldability of Sn58Bi alloy make itself easy to be shaped through LBPBF. CT pictures in

Figure 3b show that the defect bigger than 40 µm was not detected within specimens of B, C and D groups. The relative densities of specimens with 40–50 W were even higher than the relative density of as-cast alloy, and the maximum relative density (specimen C1) could reach 99.1%. On the other hand, the low vaporization temperature of the Sn58Bi alloy could cause an increase in micropores when laser molten pool temperature increases, slightly decreasing the relative density of specimen with 80 W.

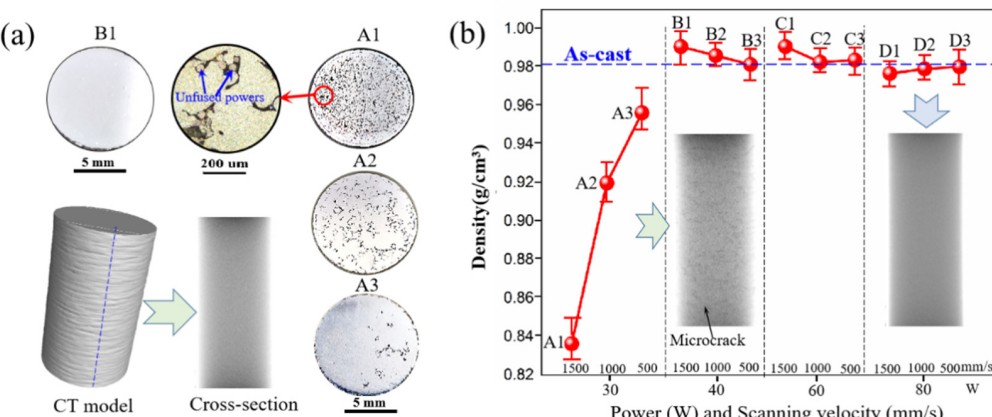

**Figure 3.** The porosity and density of LBPBF SnBi alloy specimen: (**a**) defects observed by optical microscopy and CT, (**b**) densities variation with the laser power and scanning velocity.

The microstructure was another factor which significantly affected the mechanical properties of as-printed specimens. The laser power and scanning velocity had a strong influence on the microstructure, and then on the mechanical properties of the fabricated specimens, as they were key parameters determining the solidification and cooling rates. As-printed Sn58Bi specimens fabricated in 40–80 W with fewer defects were observed by SEM, as illustrated in Figures 4 and 5. As can be seen in Figure 4, the white Bi-rich phases and black Sn-rich phases (β-Sn) constituted the microstructure of the Sn58Bi alloy. A large portion of the microstructure exhibited homogeneous irregular eutectic structures with some tiny isolated Bi-rich particles scattered within Sn-rich phase areas (see Figures 4a and 5a) when specimens were prepared at 40 W and 1500 mm/s.

For the as-cast specimen, the typical microstructure of Sn58Bi alloy consisted of regular lamellar eutectic morphology, Sn dendrite and large size faceted Bi particle, as shown in Figure 6. However, according to a previous investigation, even a small variation in the cooling rate will lead to significant changes in the microstructure and properties of SnBi alloy [23]. Therefore, the microstructure feature of the as-printed specimen could be attributed to the rapid melting–solidification process during LBPBF, during which the melt cooling rate could exceed $10^2$ K/s. During the powder melting process, with the increase in temperature, the Bi and Sn phases were melted one after another, forming a uniform liquid. During the cooling process, the β-Sn phase with a higher melting point was produced first, and the trailing Bi-phase was produced around the leading β-Sn phase, thus resulting in the formation of lamellar structure. In this case, the rapid melting–cooling rate could give rise to the convective flows, collisions between eutectic grains, the bending of primary dendrites [24] and the stirring effect of the molten pool during printing should also enhance the formation of numerous nuclei of each phase. As a result, the lamellar structure in the LBPBF specimen became less regular, their parallel arrangement deteriorated and the length therefore decreased compared to the as-cast specimen. There were also some fine Bi particles dispersed in the β-Sn matrix, which were precipitated from the β-Sn phase during the cooling process at lower temperatures. Owing to that, the solubility of Bi in the Sn phase dramatically decreased with decreasing temperature. Just as we expected, the as-printed specimen showed much better plasticity due to refined microstructure: the

crack appeared when the as-cast specimen was compressed to 60% height, while it did not appear during the compression of as-printed specimens, as shown in Figure 6.

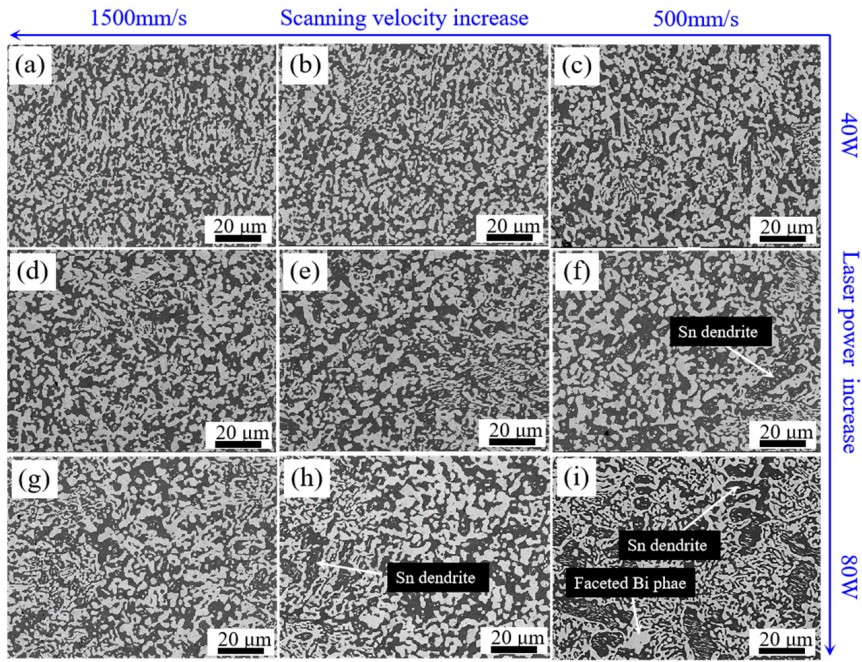

**Figure 4.** BSE images of Sn58Bi alloy specimens fabricated by LBPBF with different parameters: (**a**–**c**) 40 W with 1500, 1000 and 500 mm/s, respectively, (**d**–**f**) 60 W with 1500, 1000 and 500 mm/s, respectively, (**g**–**i**) 80 W with 1500, 1000 and 500 mm/s, respectively.

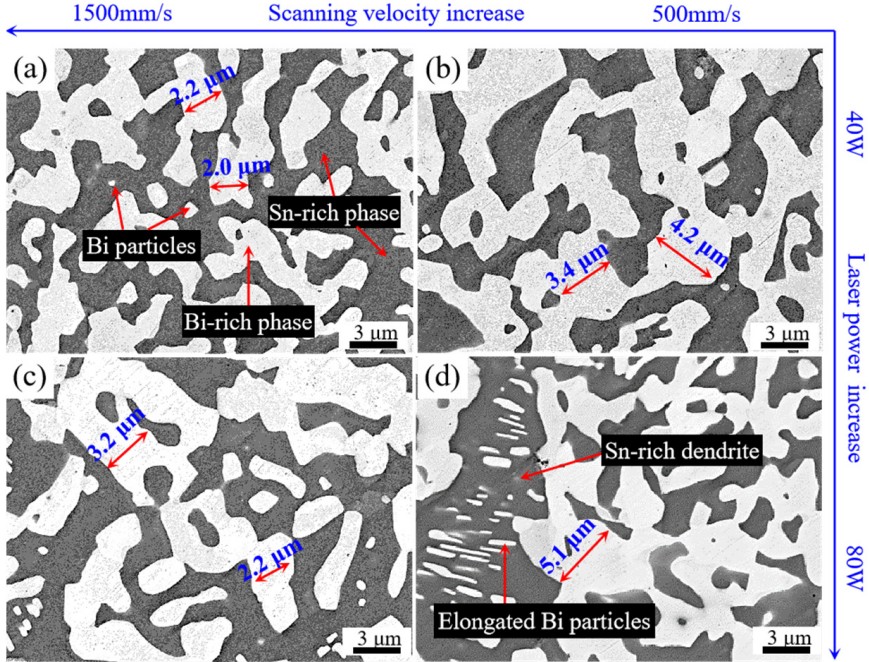

**Figure 5.** Enlarged BSE images of Sn58Bi alloy specimens: (**a**) 40 W with 1500 mm/s, (**b**) 40 W with 500 mm/s, (**c**) 80 W with 1500 mm/s, (**d**) 80 W with 500 mm/s.

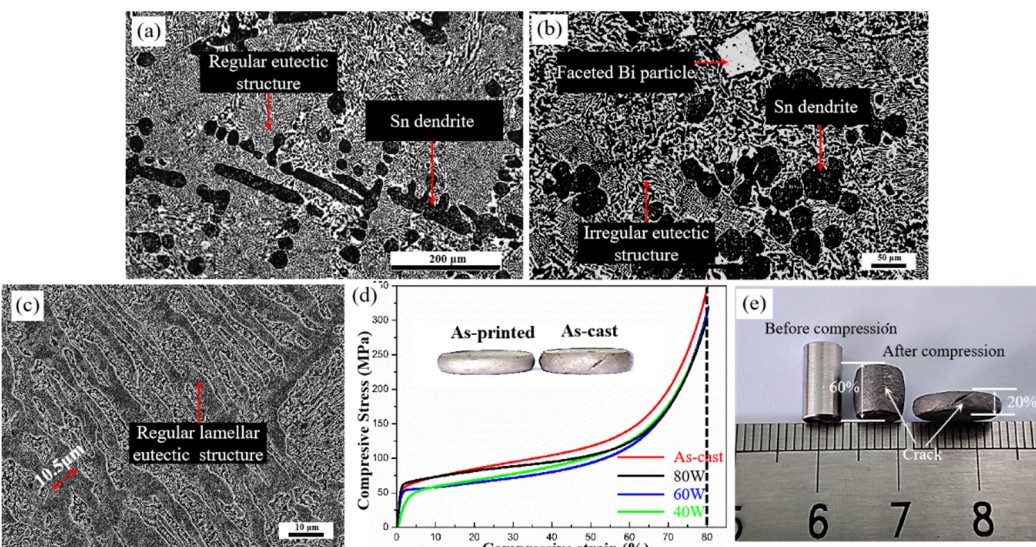

**Figure 6.** The microstructure and compression test result of as-cast Sn58Bi alloy: (**a–c**) SEM pictures of as-cast Sn58Bi alloy, (**d,e**) compression test result of as-cast and as-printed specimen.

When the laser power was increased or laser scanning speed was decreased, which indicated the increase of laser energy density, the width of the lamellar structure tended to be increased, and interphase spacing also increased gradually. As can be seen in Figure 5a,b, the maximum width of the Bi-phase in the local area evidently increased from 2.2 to 4.2 um, when the scanning velocity decreased from 1500 mm/s to 500 m/s under the laser power of 40 W. A similar situation occurred when the laser power increased from 40 to 80 W under a scanning velocity of 1500 m/s (see Figure 5a,c). At the same time, the growth of the Bi-phase made them connect to each other as reticular structures. Two reasons are probably responsible for the coarsening behavior of the Sn and Bi phases. One reason is that the cooling rate dropped significantly while the laser power increased or scanning velocity decreased, leading to the increase of residence time above the recrystallization temperature, so the grain growth time was prolonged. The more residence time of the two-phase-region provided a longer growing time for primary β-Sn, which increased the grain boundary and size of primary β-Sn [25] and provided the space for the growth of the Bi-rich phase. Another reason is that higher laser energy density enlarged the heat-affected zone, leading to the extension of the residence time at elevated temperature, which also caused the coarsening behavior of the Sn and Bi phases [7].

Therefore, when fabricated with 80 W and 500 mm/s (the highest energy density), the specimen exhibited microstructure with a large Sn-rich dendritic branch, which is similar to that of cast alloy, and a faceted Bi-rich phase with a polygon shape appeared (see Figure 4i). It could be speculated that high scanning energy density also prolonged the growth time for primary β-Sn before the Bi-rich phase appeared so as to leave enough time for the formation of a large size Sn-rich large dendritic branch. On the contrary, the similar width of the Sn-rich and Bi-rich phases in low energy density indicated the coupled growth of the two phases (see Figure 4). The elongated Bi particles could be found within Sn-rich dendritic branch area, as shown in Figure 5d, indicating the lower cooling rate.

For the two phases of Sn58Bi alloy, the Bi-phase behaves with relatively higher strength and indicates a strengthening effect. Because the volumes of the two phases were relatively close in Sn58Bi alloy, and the Sn phase cannot be connected, the two phases deform with approximately equal strain, and the higher strength Bi-phase plays a critical role in deformation. The lamellar Bi-phase works as the support skeleton in the deformation. With the increase of laser power, the Bi-phase became coarsened and cross-linked, which improved the support effect.

In order to study the influence of LBPBF parameters on the composition of the matrix, the mass fraction of the Sn element in the β-Sn phase was measured by EDS, and three points were randomly selected in each specimen, as depicted in Figure 7a. The results in Figure 7b manifested that the average mass fraction of the Sn element in the β-Sn phase of each specimen had no notable regular change with the variation of printing parameters, which was probably owing to the uneven composition distribution in the local area of the β-Sn matrix, but the average mass fractions were nearly same and mainly distributed between 97% and 98%.

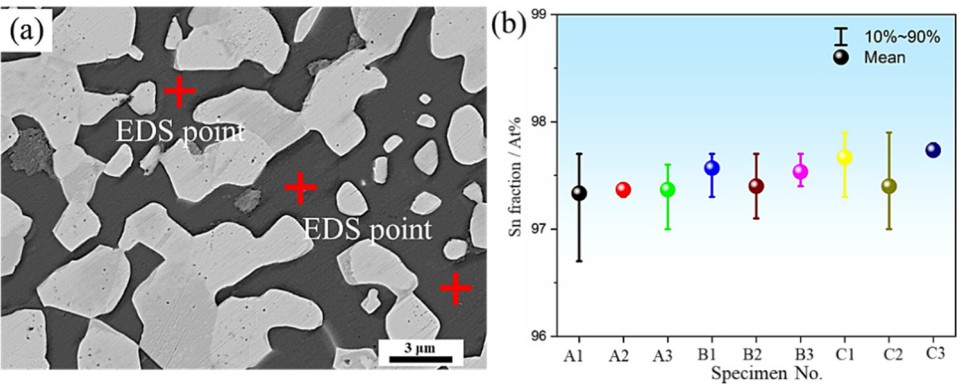

**Figure 7.** The EDS points measurement in β-Sn matrix of printed Sn58Bi specimens: (**a**) measurement locations, (**b**) Sn fraction.

Figure 8 show the XRD results for the as-printed Sn58Bi alloy and Sn58Bi alloy powders. The position and intensity of XRD peaks indicated there were no apparent phase differences between specimens fabricated in different LBPBF parameters, just as shown in Figure 8a. However, by comparing several peaks in the magnified XRD pattern picture of Figure 8b, the peaks broadened and weakened with the increase in laser power or decrease in scanning velocity. For specimens printed with low energy density, such as 40 W laser power and 1500 mm/s scanning velocity, the XRD peaks were inclusive in shape, similar to Sn58Bi alloy powders. However, the XRD peaks became so smooth and widened that the XRD peaks near 45° were even merged for specimens printed with laser energy density as high as 80 W. The broad effect of XRD peaks usually could be ascribed to two reasons: finer grain size or residual stress. However, microstructure pictures in Figure 4 indicate that grain coarsening rather than grain refinement occurred with the increase of printing laser energy density. After obviating the likelihood of solid solubility variation of Bi in β-Sn, which could also cause the shifting of XRD peaks, the shifting and weakening of peaks was another proof of the existence of residual stress. The peak intensity weakening was very evident for specimens printed with 500 mm/s, which was related to residual stress due to uneven cooling from high temperatures. As we know, the residual stress in specimens could increase their strength, so strength improvement was more obvious as laser energy density increased.

The dimension accuracy and surface quality are other standards of evaluating the additive manufacturing specimens. The increment of laser power and decrement of scanning velocity tended to increase the clad width (as the melt pool size increases) and the height (up to a certain value). As shown in Figure 9, the top surface of the specimen, which was printed in low energy density (40 W and 1500 mm/s), was concave, but as the laser energy density increased, the top surface gradually became convex and smooth. At the same time, the numbers printed on the top surface became more blurred for high energy density specimens (80 W and 500 mm/s), indicating a reduction in dimension accuracy. The side view of the specimen was conducted as shown in Figure 9b, and there were no obvious differences among the side surfaces of the three specimens. Measurement by vernier caliper indicated that the actual height of specimens B1, C2 and D3 was 9.98, 9.96

and 9.87 mm, when the assumed heights of the specimen were 10.00 mm. So high energy density specimen has the lowest dimensional accuracy in height, which may be due to the convex surface and high stress, which make powder hard to spread on the top surface.

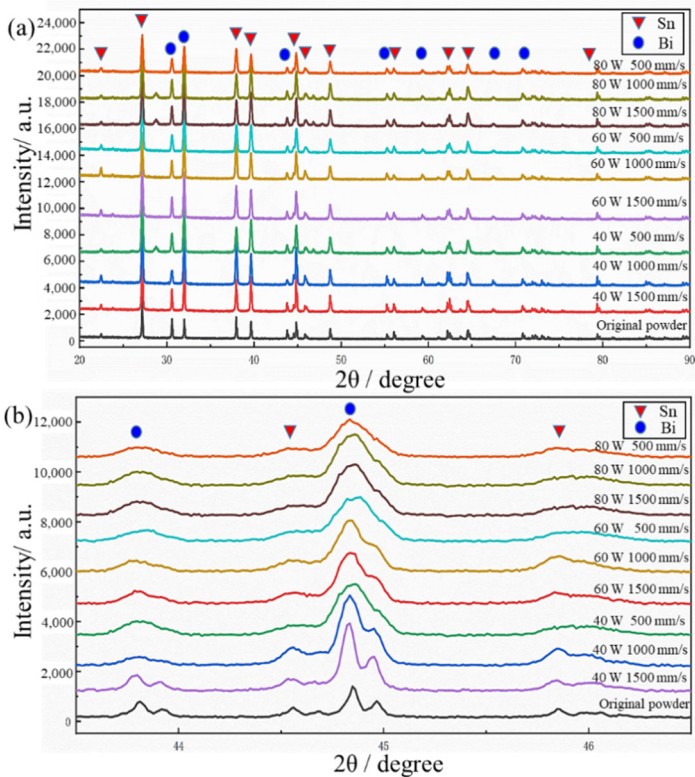

**Figure 8.** XRD patterns of the as-printed Sn58Bi alloy and Sn58Bi alloy powder: (**a**) 20–90° diffraction angle, (**b**) 43–47° diffraction angle.

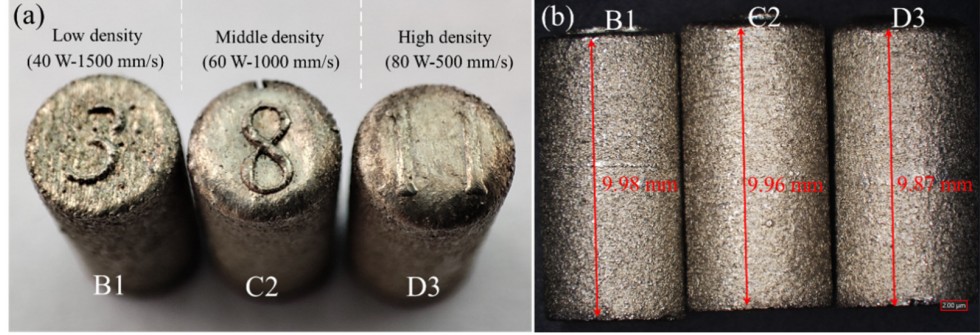

**Figure 9.** Surface of the as-printed Sn58Bi alloy specimens in different energy densities: (**a**) top view, (**b**) side view.

## 4. Conclusions

In summary, the LBPBF was adopted to the fabricated Sn58Bi alloy specimens in different parameters. The microstructure and mechanical properties were studied, and the main conclusions are drawn as follows:

(1) For the as-cast specimen, the typical microstructure of Sn58Bi alloy consisted of regular lamellar eutectic morphology, Sn dendrite and large size faceted Bi particle. However, for the as-printed specimen, the lamellar structure in the LBPBF specimen became less regular, their parallel arrangement deteriorated, and the length, therefore, decreased, compared to the as-cast specimen. There were also some fine Bi particles rather

than large size faceted Bi particles dispersed in the β-Sn matrix. Comparing the Bi-phase size of as-cast and as-printed specimens, the Bi-phase of the as-cast specimen was much coarser than the Bi-phase of the as-printed specimen. Just as we expected, the as-printed specimen showed much better plasticity due to its refined microstructure: a crack appeared when the as-cast specimen was compressed to 60% height, while it did not appear during the compression of as-printed specimens.

(2) The Sn58Bi alloy showed good formability when specimens were prepared by LBPBF between 40–80 W. However, under the laser power lower than 40 W, defects were prone to appear in the interior of specimens, resulting in low density and strength. Under high laser power (higher than 40 W), CT pictures could only show that a defect bigger than 40 μm was not detected in specimens of B, C and D groups. The low melting point (138 °C) and excellent weldability of Sn58Bi alloy make it easy to be shaped through LBPBF. The relative densities of specimens with 40–50 W were even higher than the relative density of as-cast alloy, and the maximum relative density (specimen C1) could reach 99.1%. On the other hand, the low vaporization temperature of Sn58Bi alloy could cause an increase in micropores when laser molten pool temperature increases, slightly decreasing the relative density of specimen with 80 W. Therefore, laser power of 40–60 W is more helpful to obtain a higher density of as-printed alloy.

(3) In the laser power range of 40–80 W, with the variation of laser power or scanning velocity, the laser energy density changed accordingly and the mechanical strengths of specimens were improved with the increase of laser power density. This strength change may be related to the microstructure evolution and internal residual stress in the printing process. For the two phases of Sn58Bi alloy, the Bi-phase behaves with relatively higher strength and indicates a strengthening effect. Because the volumes of the two phases were relatively close in the Sn58Bi alloy, and the Sn phase cannot be connected with each other, thereby the two phases deform with approximately equal strain, and the higher strength Bi-phase plays a critical role in deformation. The lamellar Bi-phase will work as the support skeleton in the deformation. With the increase of laser power, the Bi-phase becomes coarsened and cross-linked, which improves the support effect. On the other hand, with the increase of laser power, the cooling rate during LBPBF increases leading to the increase of residual stress; thus, the specimen will also be strengthened.

(4) Compared to the as-cast specimen, the as-printed specimen presents a more refined structure, so they also show better plasticity. Although the high-power structure is close to the as-cast structure, the structure is more refined, and the Bi-phase presents irregular distribution, leading to higher yield strength.

(5) The power laser density also has a significant impact on surface quality and dimensional accuracy. Poor surface quality easily appears at low laser power. As the laser energy density increases, the top surface gradually becomes convex and smooth, but dimension accuracy decreases.

**Author Contributions:** Conceptualization, C.Y.; methodology, C.Y., K.S.; data curation, C.Y., Y.W., D.D. and B.Z.; writing—original draft preparation, C.Y.; writing—review and editing, Y.Z., M.S., J.C. and Y.W.; supervision, Y.Z.; project administration, Y.Z.; funding acquisition, C.Y. All authors have read and agreed to the published version of the manuscript.

**Funding:** This research was funded by the National Natural Science Foundation of China (C.Y.: 52105410).

**Institutional Review Board Statement:** Not applicable.

**Informed Consent Statement:** Not applicable.

**Data Availability Statement:** The raw/processed data required to reproduce these findings cannot be shared at this time as the data also form part of an ongoing study.

**Conflicts of Interest:** The authors declare no conflict of interest.

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
