# Peer review of "Microstructure and Mechanical Properties of Sn58Bi Components Prepared by Laser Beam Powder Bed Fusion"

_metals, doi:10.3390/met12071135_

Round 1

Reviewer 1 Report

Dear authors,

the expansion of the range of materials for AM processes and the widening of the knowledge of suitable fabrication parameters for these is highly beneficial for the research community and industry alike. With this, the topic the manuscript touches on is definitely of interest.

However, there are quite some points I do need to criticize about the submitted manuscript: #

- vague descriptions are being made throughout the entire manuscript which do not add any merit to the work. For example, page 1 line 43: “more or less”; page 4 line 114: “were so good”; page 6 line 188: “mushy zone”

- addition of alloying elements (third elements) is mentioned as improvement in the abstract and in the introduction, but is not part of the undertaken study.

- SLM is an outdated description and became a trademark for SLM Solitions GmbH. These days the agreed standard should be employed as proper nomenclature, see ISO/ASTM 52900:2021

- pag e2 line 47: Heterogeneous nucleation sites may be important for cast, but not in the PBF process, where solidification occurs rapidly from the melt pool boundaries

- page 2 lines 50 to 62: correlation to the work at hand is unclear. Why is it better to not use one of those alloys mentioned there and go with Sn58Bi instead? Is this the first study to investigate Sn58Bi and is there preliminary knowledge about its processability?

- general comment: parameters to be written in italic

- page 3: alloy composition is only provided for SLM samples, but not for casted samples; Process parameters are not comprehensive, e.g. layer thickness, hatch distance and possible preheating are not disclosed

- Methods chapter is incomplete. Casting process, as well as ALL further investigations (material testing, CT scans, …) are not described properly.

- Page 3 line 106: what is the content of the earlier exploration experiments? If there are preliminary results, why weren’t these addressed in the introduction??

- relative density is an important measure for the sample quality and should be amongst the first results to be discussed. Also, no real evaluation (e.g. relative density correlation to parameter settings) is done to judge about the suitable process window.

- page 4 line 132: “CT pictures proved”   …… well, there is no proper evaluation of the CT data which would allow to do so

- page 5: phases and phase distribution is shown, but it was not addressed if this microstructure was expected and whether or not it is beneficial for this alloy type. Later in the conclusion it only is mentioned that it is more refined, but it remains unclear how this is to be judged. One could add information about grain size differences, the grain shape and their impact in the comparison of cast vs. SLM

- General comment: discussion fails to communicate whether the results were desirable and if they are reasonable (in comparison to other studies).

- Page 9 line 250: CT could not detect internal defects. This statement is meaningless, since the resolution of the CT analyses was not mentioned. E.g. if the voxel size is too large to display cracks, pores, …

To sum up, the manuscript is written in a vague manner and arguments made are often inconclusive. The comparison of SLM and cast is lacking, as no real comparison was made. Results cannot be reproduced as important information is not provided. Results were not properly evaluated and discussed and hence, no real key takeaways could be presented.

Author Response

  1. Vague descriptions are being made throughout the entire manuscript which do not add any merit to the work. For example, page 1 line 43: “more or less”; page 4 line 114: “were so good”; page 6 line 188: “mushy zone”.

Response to the comment:

Thanks a lot for your suggestion. According to your advice, we have checked the entire manuscript and made corresponding revision in the paper.

  1. Addition of alloying elements (third elements) is mentioned as improvement in the abstract and in the introduction, but is not part of the undertaken study.

Response to the comment:

Thanks a lot for your comments. According to the previous study, the brittleness of Sn58Bi alloy is a limitation that has to be addressed for its wide applications [7]. Brittleness of SnBi alloy is mostly caused by the intrinsic brittle Bi phase and coarsened microstructure [8]. In fact, two methods had been adopted to solve this problem: addition of alloying elements and severe plastic deformation. And their mechanical properties improvement mechanisms were both due to microstructure refinement effect. However, the addition of alloying elements may change the melting point of solder paste alloy, while severe plastic deformation methods, such as equal channel angular pressing process (ECAP) and high pressure torsion (HPT) [14-15], were not suitable for the large bulk materials. So a new approach to improve mechanical performance without changing melting point of SnBi alloy is urgently needed. According to previous investigation, even small variation in the cooling rate will lead to significant changes in microstructure and properties of SnBi alloy [16], and rapid cooling process probably lead to the fine structure of SnBi alloy. As we known, the extremely rapid solidification and cooling process were the unique characteristics of most AM technology, such as LBPBF. Thus, LBPBF technology was firstly tried to fabricate Sn58Bi specimens in order to get fine structure and better mechanical performance.  

We have made corresponding revision in introduction section in order to make the paper more logical and clearer, which can be found on Page 1-2 in the section of Introduction of the revised manuscript, highlighted in yellow color.

Relevant references:

[7] Zhou S., Shen Y.A., Tiffani U., Vasanth C.S., etc. Improve mechanical properties induced by In and In&Zn double additions to eutectic Sn58Bi alloy, J. Mater. Sci. –Mater. Electron. 2019, 30, 7423-7434.

[8] L.Yang, W. Zhou, Y. Ma, X. Li, Y. Liang, W. Cui, P. Wu, Effects of Ni addition on mechanical properties of SnBi solder alloy during solid-state aging. Mater. Sci. Eng. A, 2016, 633, 367-375.

[14] Wang, C. T. , He, Y. , & Langdon, T. G. . The significance of strain weakening and self-annealing in a superplastic Bi-Sn eutectic alloy processed by high-pressure torsion. Acta Mater. 2019, 185, 245-256.

[15] Alkorta, J., Sevillano, J. Measuring the strain rate sensitivity by instrumented indentation. Application to an ultrafine grain (equal channel angular pressed) eutectic Sn–Bi alloy. J. Mater. Res., 2004, 19, 282-290.

[16] O.V. Gusakova, V.G. Sphephelevich, L.P. Shcherbachenko, Influence of the melt cooling rate on the microstructure and texture of Sn-Bi and Sn-Pb eutectic alloys, J. Surf. Invest. -X-RAY+, 2016, 10,146-152.

  1. SLM is an outdated description and became a trademark for SLM Solitions GmbH. These days the agreed standard should be employed as proper nomenclature, see ISO/ASTM 52900:2021

Response to the comment:

Thank you very much for your comments. We have revised the paper and replaced selective laser melting (SLM) with laser beam powder bed fusion (LBPBF).  

  1. Page 2 line 47: Heterogeneous nucleation sites may be important for cast, but not in the PBF process, where solidification occurs rapidly from the melt pool boundaries.

Response to the comment:

Many thanks for your comments. We are totally agree with you, heterogeneous nucleation sites were important for cast rather than the PBF process. In the introduction section, the heterogeneous nucleation sites were used to explain the reason that microstructure refinement of as-cast Sn58Bi alloy. As-cast state of Sn58Bi alloy was declared in the corresponding paragraph of revised manuscript.

  1. Page 2 lines 50 to 62: correlation to the work at hand is unclear. Why is it better to not use one of those alloys mentioned there and go with Sn58Bi instead? Is this the first study to investigate Sn58Bi and is there preliminary knowledge about its processability?

Response to the comment:

Many thanks for your comments. We have been studied the as-cast SnBi alloy for mangy years, and the brittleness of Sn58Bi alloy is a limitation that has to be addressed for the wide applications [7]. Brittleness of SnBi alloy is mostly caused by the intrinsic brittle Bi phase and coarsened microstructure [8]. In fact, two methods had been adopted to solve this problem: addition of alloying elements and severe plastic deformation. And their mechanical properties improvement mechanisms were both due to microstructure refinement effect. However, the addition of alloying elements may change the melting point of solder paste alloy, while severe plastic deformation methods, such as equal channel angular pressing process (ECAP) and high pressure torsion (HPT) [14-15], were not suitable for the large bulk materials. So a new approach to improve mechanical performance without changing melting point of SnBi alloy is urgently needed. According to previous investigation, even small variation in the cooling rate will lead to significant changes in microstructure and properties of SnBi alloy [22], and rapid cooling process probably lead to the fine structure of SnBi alloy. As we known, the extremely rapid solidification and cooling process were the unique characteristics of of most AM technology, such as LBPBF. Thus, LBPBF technology was firstly tried to fabricate Sn58Bi specimens in order to get fine structure and better mechanical performance.

  1. General comment: parameters to be written in italic?

Response to the comment:

Many thanks for your comments. We have made corresponding revision in the revised manuscript.

  1. Page 3: alloy composition is only provided for SLM samples, but not for casted samples; Process parameters are not comprehensive, e.g. layer thickness, hatch distance and possible preheating are not disclosed?

Response to the comment:

Many thanks for your comments. Alloy composition for as-cast samples was added in Materials and Methods section, and process parameters including layer thickness and hatch distance were also added. We have made corresponding revision in the page 3 of revised manuscript, highlighted in yellow color.

Layer thickness were set 60um, and hatch distance were set 0.05mm. Laser spot diameter was 0.06mm. Preheating was not used before printing.

Table 1. Chemical compositions (wt %) of the Sn58Bi powder used and as-cast Sn58Bi alloy

Condition

Sn

Bi

In

Pb

Ge

Sb

O

Ag

Power

Bal.

57.86

0.011

0.007

0.006

0.005

0.005

0.002

As-cast

Bal.

57.83

0.015

0.005

0.005

0.004

0.005

0.003

  1. Methods chapter is incomplete. Casting process, as well as ALL further investigations (material testing, CT scans, …) are not described properly.

Response to the comment:

Many thanks for your comments. We have made corresponding revision in page 3-4 of the revised manuscript, highlighted in yellow color.

The Sn58Bi alloy pieces were melted and heated to 250°C in smelting furnace, and then the molten metal was poured into sand mold at 190-200°C. After cooling to room temperature and cutting riser, the Sn588Bi alloy bars (ø30×150 mm) were obtained, which could be machined to the compression test specimens.

The computed tomography (CT) was employed to scan defects within specimens by using micro-CT. X-Ray-Tube voltage was 200KV, X-Ray-Tube current was 0.11mA, magnification was 11.31, 3D-Pixel size was 0.010mm, and resolution was about 40μm. After the surfaces of column specimens were polished by sand papers, the uniaxial compression test was conducted on an Instron 5985 test machine with a compressing speed of 0.5 mm/min under ambient temperature, so as to investigate the influence of LBPBF parameters on mechanical properties of different specimens.

  1. Page 3 line 106: what is the content of the earlier exploration experiments? If there are preliminary results, why weren’t these addressed in the introduction??

Response to the comment:

Many thanks for your comments. We are sorry to confuse you. Besides preparation with power of 30W-80W, 20W and 90W were also tried for specimen printing in this study. However, Sn58Bi alloy power could not be melt sufficiently and bonded into bulk material below laser power of 30W at any scanning velocity (see Fig. 2a), while the heavy dark smoke appeared above laser power of 80W,which may lead to the failure of scanning mirror. So the laser powers between 30W and 80W were adopted to prepare specimens in order to study their mechanical properties and microstructure.

We have made corresponding revision in page 3 of the revised manuscript, highlighted in yellow color.

Figure 2. LBPBF Sn58Bi specimens (a) and uniaxial compression test results (b).

  1. Relative density is an important measure for the sample quality and should be amongst the first results to be discussed. Also, no real evaluation (e.g. relative density correlation to parameter settings) is done to judge about the suitable process window.

Response to the comment:

Many thanks for your comments. Relative density variation with the change of printing parameters was used to replace real density and illustrated in Figure 3. The relative densities of as-printed specimens fabricated with 30W laser power were ranged from 85.0% to 93.8%, which were much smaller than other specimens. The optical pictures of cross-section in Fig. 3a showed that a mass of voids were easily observed on the surface, and the CT pictures also illustrated a mass of defects (dark shadow) existed in the specimens, which indicated that specimens had high porosity under low laser power. In fact, the voids were caused by unfused power under low laser power (see Fig. 3a), leading to the low relative density. When the laser power was higher than 40W, the relative densities of specimens dramatically increased to the value around 98%. As one of solder paste materials, the low melting point (138°C) and excellent weldability of Sn58Bi alloy make itself easy to be shaped through LBPBF. CT pictures in Figure 3b showed that the defect bigger than 40μm was not detected within specimens of B, C and D group. The relative densities of specimens with 40-50W were even higher than relative density of as-cast alloy, and the maximum relative density (specimen C1) could reach 99.1%. On the other hand, the low vaporization temperature of Sn58Bi alloy could cause an increase in micropores when laser molten pool temperature increase, slightly decreasing the relative density of specimen with 80W. Therefore, it could be considered that laser power of 40-60W is more helpful to get higher density of as-printed alloy.

We have made corresponding revision in page 3-4 of the revised manuscript, highlighted in yellow color.

Figure 3. The porosity and density of LBPBF SnBi alloy specimen: (a) defects observed by optical microscopy and CT, (b) densities variation with the laser power and scanning velocity.

  1. Page 4 line 132: “CT pictures proved”   …… well, there is no proper evaluation of the CT data which would allow to do so

Response to the comment:

Many thanks for your comments. We are totally agree with you, CT pictures could only show that the defect bigger than 40μm was not detected in specimens of B, C and D group, since the resolution of micro-CT was 40μm. We have made corresponding revision in page 3 of the revised manuscript, highlighted in yellow color.

  1. Page 5: phases and phase distribution is shown, but it was not addressed if this microstructure was expected and whether or not it is beneficial for this alloy type. Later in the conclusion it only is mentioned that it is more refined, but it remains unclear how this is to be judged. One could add information about grain size differences, the grain shape and their impact in the comparison of cast vs. SLM.

Response to the comment:

Thank you for your comments. For as-cast specimen, the typical microstructure of Sn58Bi alloy consisted of regular lamellar eutectic morphology as shown in Fig.6, and Sn dendrite and large size faceted Bi particle as shown in Fig. 6b. But for the as-printed specimen, the lamellar structure in LBPBF specimen became less regular, their parallel arrangement deteriorates, and the length therefore decrease, compared to as-cast specimen. There were also some fine Bi particles rather than large size faceted Bi particle dispersed in the β-Sn matrix as shown in Figure 4a-h. Comparing the Bi phase size of as-cast and as-printed specimens, the Bi phase of as-cast specimen in Figure 6c was much coarser than Bi phase of as-printed specimen in Figure 5a-d.

Figure 4. BSE images of Sn58Bi alloy specimens fabricated by LBPBF with different parameters: (a-c) 40W with 1500, 1000 and 500 mm/s respectively, (d-f) 60W with 1500, 1000 and 500 mm/s respectively, (g-i) 80W with 1500, 1000 and 500 mm/s respectively.

Figure 5. Enlarged BSE images of Sn58Bi alloy specimens: (a) 40W with 1500 mm/s, (b) 40W with 500 mm/s, (c) 80W with 1500 mm/s, (c) 80W with 500 mm/s.

Figure 6. The microstructure and compression test result of as-cast Sn58Bi alloy:

(a-c) SEM pictures of as-cast Sn58Bi alloy, (d-e) compression test result of as-cast and as-printed specimens.

According to previous investigation, even small variation in the cooling rate will lead to significant changes in microstructure and properties of SnBi alloy [16], and rapid cooling process probably leading to the fine structure of SnBi alloy. This is the reason why we adopted LBPBF to prepare Sn58Bi specimen. So the microstructure feature of as-printed specimen could be attributed to the rapid melting-solidification process during LBPBF, during which melt cooling rate could excess 102 K/s. Just as we expect, the as-printed specimen showed much better plasticity due to refine microstructure: the crack appeared when as-cast specimen was compressed to 80% height, while it didn’t appear during the compression of as-printed specimens.

Relevant references:

  • V. Gusakova, V.G. Sphephelevich, L.P. Shcherbachenko, Influence of the melt cooling rate on the microstructure and texture of Sn-Bi and Sn-Pb eutectic alloys, J. Surf. Invest. -X-RAY+, 2016, 10,146-152.

  1. General comment: discussion fails to communicate whether the results were desirable and if they are reasonable (in comparison to other studies).

Response to the comment:

Many thanks for your comments. According to previous investigation, even small variation in the cooling rate will lead to significant changes in microstructure and properties of SnBi alloy [16], and rapid cooling process probably leading to the fine structure of SnBi alloy. This is the reason why we adopted LBPBF to prepare Sn58Bi specimen. So the microstructure feature of as-printed specimen could be attributed to the rapid melting-solidification process during LBPBF, during which melt cooling rate could excess 102 K/s. Just as we expect, the as-printed specimen showed much better plasticity due to refine microstructure: the crack appeared when as-cast specimen was compressed to 60% height, while it didn’t appear during the compression of as-printed specimens as shown in Figure 6. 

We have made corresponding revision in discussion part of the revised manuscript, highlighted in yellow color.

Relevant references:

[16] O.V. Gusakova, V.G. Sphephelevich, L.P. Shcherbachenko, Influence of the melt cooling rate on the microstructure and texture of Sn-Bi and Sn-Pb eutectic alloys, J. Surf. Invest. -X-RAY+, 2016, 10,146-152.

  1. Page 9 line 250: CT could not detect internal defects. This statement is meaningless, since the resolution of the CT analyses was not mentioned. E.g. if the voxel size is too large to display cracks, pores.

Response to the comment:

Many thanks for your comments. We are totally agree with you, CT pictures could only show the defect bigger than 40μm was not detected in specimens of B, C and D group, since the resolution of micro-CT was 40μm. We have made corresponding revision in the revised manuscript, highlighted in yellow color.

Reviewer 2 Report

Manuscript numbered “metals-1765791” has been reviewed:

The introduction needs some improvements.

Please add a suitable scale bar for figures.

Please use laser beam powder bed fusion (LBPBF) instead of selective laser melting (SLM)

Please use a unique naming procedure, what is the process parameter for samples shown in figure 9?

Results have been just reported, please compare your finding with other research.

There is no information about the compression test.

Add reasons and mechanisms for the relation between process parameters and mechanical properties and density.

Compare the result of the LBPBF method with other additive manufacturing methods.

Fallowing papers are suggested for the introduction and result section:

Build position-based dimensional deviations of laser powder-bed fusion of stainless steel 316L

Flow field analysis for multilaser powder bed fusion and the influence of gas flow distribution on parts quality

High-cycle fatigue properties of curved-surface AlSi10Mg parts fabricated by powder bed fusion additive manufacturing

The influence of laser power and scanning speed on the microstructure and surface morphology of Cu2O parts in SLM

A critical review of 3D printing and digital manufacturing in construction engineering

Author Response

Thank you very much for your comprehensive comments and thoughtful suggestions on our manuscript. They are very helpful for us to improve the quality of the manuscript. According to these comments and suggestions, we have carefully made modifications on the original manuscript (Manuscript ID metals-1765791).

  1. Please add a suitable scale bar for figures.

Response to the comment:

 Many thanks for your comments. We have made corresponding revision in Fig. 2 and 3 of the revised manuscript.

  1. Please use laser beam powder bed fusion (LBPBF) instead of selective laser melting (SLM).

Response to the comment:

Thanks a lot for your suggestion. We have made corresponding revision in the revised manuscript.

  1. Please use a unique naming procedure, what is the process parameter for samples shown in figure 9?

Response to the comment:

Thanks a lot for your comments. We are sorry to confuse you. The specimen with 40W and 1500mm/s was chosen as low energy density sample, the specimen with 60W and 1000mm/s was chosen as middle energy density sample, and the specimen with 80W and 500mm/s was chosen as high energy density sample. We have made corresponding revision in the revised manuscript, highlighted in yellow color.

  1. There is no information about the compression test.

Response to the comment:

Thanks a lot for your comments. After the surfaces of column specimens were polished by sand papers, the uniaxial compression test was conducted on an Instron 5985 test machine with a compressing speed of 0.5 mm/min under ambient temperature, so as to investigate the influence of LBPBF parameters on mechanical properties of different specimens. We have made corresponding revision in the revised manuscript, highlighted in yellow color.

  1. Add reasons and mechanisms for the relation between process parameters and mechanical properties and density.

Response to the comment:

Thanks a lot for your suggestions. Relative density variation with the change of printing parameters was illustrated in Figure 3. The relative densities of as-printed specimens fabricated with 30W laser power were ranged from 85.0% to 93.8%, which were much smaller than other specimens. The optical pictures of cross-section in Fig. 3a showed that a mass of voids were easily observed on the surface, and the CT pictures also illustrated a mass of defects (dark shadow) existed in the specimens, which indicated that specimens had high porosity under low laser power. In fact, the voids were caused by unfused power under low laser power (see Fig. 3a), leading to the low relative density. When the laser power was higher than 40W, the relative densities of specimens dramatically increased to the value around 98%. As one of solder paste materials, the low melting point (138°C) and excellent weldability of Sn58Bi alloy make itself easy to be shaped through LBPBF. CT pictures in Figure 3b showed that the defect bigger than 40μm was not detected within specimens of B, C and D group. The relative densities of specimens with 40-50W were even higher than relative density of as-cast alloy, and the maximum relative density (specimen C1) could reach 99.1%. On the other hand, the low vaporization temperature of Sn58Bi alloy could cause an increase in micropores when laser molten pool temperature increase, slightly decreasing the relative density of specimen with 80W. Therefore, it could be considered that laser power of 40-60W is more helpful to get higher density of as-printed alloy.

A large portion of microstructure exhibited homogeneous irregular eutectic structures with some tiny isolated Bi-rich particles scattered within Sn-rich phase areas (see Fig. 4a and 5a) when specimens were prepared in 40W and 1500mm/s. When the laser power was increased or laser scanning speed was decreased, which indicated the increase of laser energy density, the width of lamellar structure tended to be increased and interphase spacing also increased gradually. As can be seen in Fig. 5a and 5b, the maximum width of Bi phase in local area evidently increased from 2.2um to 4.2um, when the scanning velocity decreased from 1500mm/s to 500m/s under the laser power of 40W. The similar situation occurred when the laser power increased from 40W to 80W under scanning velocity of 1500m/s (see Fig. 5a and 5c). At the same time, the growth of Bi phase made themselves connect each other as reticular structure. Two reasons are probably responsible for coarsening behavior of Sn and Bi phases. One reason is that the cooling rate dropped significantly while the laser power increased or scanning velocity decreased, leading to the increase of residence time above recrystallization temperature, so the grain growth time was prolonged. The more residence time of two-phase-region provided longer growing time for primary β-Sn, which increased the grain boundary and size of primary β-Sn [27] and provided the space for the growth of Bi-rich phase. Another is that higher laser energy density enlarged heat affected zone, leading to extension of residence time at elevated temperature, which also could cause coarsening behavior of Sn and Bi phases [7].

For the two phases of Sn58Bi alloy, Bi phase behaves with relatively higher strength and indicates strengthening effect. Because the volumes of two phases were relatively close in Sn58Bi alloy, and the Sn phase cannot be connected with each other, thereby the two phases deform with approximately equal strain and the higher strength Bi phase plays a critical role in deformation. The lamellar Bi phase will work as the support skeleton in the deformation. With the increase of laser power, the Bi phase became coarsened and cross-linked, which improved the support effect.

We have made corresponding revision in the revised manuscript.

Relevant references:

[7] Zhou S., Shen Y.A., Tiffani U., Vasanth C.S., etc. Improve mechanical properties induced by In and In&Zn double additions to eutectic Sn58Bi alloy, J. Mater. Sci. –Mater. Electron. 2019, 30, 7423-7434.

[21] Yang Liu, Haifeng Fu, Fenglian Sun, etc. Microstructure and mechanical properties of as-reflow Sn58Bi composite solder paste, J. Mater. Process. Tech., 2016, 238, 290-296.

  1. Compare the result of the LBPBF method with other additive manufacturing methods.

Fallowing papers are suggested for the introduction and result section:

Build position-based dimensional deviations of laser powder-bed fusion of stainless steel 316L

Flow field analysis for multilaser powder bed fusion and the influence of gas flow distribution on parts quality

High-cycle fatigue properties of curved-surface AlSi10Mg parts fabricated by powder bed fusion additive manufacturing

The influence of laser power and scanning speed on the microstructure and surface morphology of Cu2O parts in SLM

A critical review of 3D printing and digital manufacturing in construction engineering

Response to the comment:

Thanks a lot for your suggestions. We have added the references in the revised manuscript, highlighted in yellow color.

Additive manufacturing technology can be used to fabricate components with complex structure directly and rapidly [17]. LBPBF provides improvements in product quality, processing time, and manufacturing reliability compared to binder-based laser sintering AM processes. In fact, the stainless steel 316L, AlSi10Mg, Cu2O and many other materials had been used to fabricate components by using LBPBF technology, and the influence of parameters on the microstructure and properties was also investigated by researchers[18-21].

Relevant references:

[17] Wang L,; Wei Q. S.; Xue P. J.; et al. Fabricate Mould Insert with Conformal Cooling Channel Using Selective Laser Melting. Adv. Mater.s Res. 2012, 502, 67-71.

[18] Jithin K.V., Mahyar K., Amir H.G. , et al. (2021). Build position-based dimensional deviations of laser powder-bed fusion of stainless steel 316l - sciencedirect. Precis. Eng., 2020, 67, 58-68.

[19] Liu, Z., Yang, Y., Wang, D., Chen, J., Xiao, Y., Zhou, H., Chen, Z. and Song, C.  Flow field analysis for multilaser powder bed fusion and the influence of gas flow distribution on parts quality, Rapid Prototyping J. 2022, ahead-of-print https://doi.org/10.1108/RPJ-12-2021-0351

[20] Zhou, Y., Abbara, E.M., Jiang, D., Azizi, A., Poliks, M.D. and Ning, F. High-cycle fatigue properties of curved-surface AlSi10Mg parts fabricated by powder bed fusion additive manufacturing, Rapid Prototyping J., 2022, ahead-of-print.  https://doi.org/10.1108/RPJ-09-2021-0253

[21] Abid Ullah, Asif Ur Rehman, Metin Uymaz Salamci, Fatih Pıtır, Tingting Liu, The influence of laser power and scanning speed on the microstructure and surface morphology of Cu2O parts in SLM, Rapid Prototyping J., 2022, ahead-of-print, https://doi.org/10.1108/RPJ-12-2021-0342

Reviewer 3 Report

Dear Authors,

The main goal of the paper is the evaluation of SLM 3D printing parameters like scanning velocity and laser power on the microstructure and mechanical properties of Sn58Bi material samples. The main idea of carried out work seems to be interesting nevertheless, there are a few issues that have to be improved in the paper before its publishing.
1)    Introduction.
There is no clear justification why the Sn58Bi material should be implemented in Additive Manufacturing techniques and what kind of benefits can be achieved using this material in a 3D printing process. Furthermore, can you give more details about the implementation of this material in particular branches of the industry where AM technique was used? It allows better defining the scientific background of your work and better justified its importance.
2)    Figure 1 – there is a lack of a chart illustrating the powder diameter distribution
3)    The quality of the figures is very low, please put them in a higher resolution.
4)    Line 91 there is information that three different values of the scanning velocity and laser power were applied in the studies. I understand that this approach gave a 3x3 matrix of material samples, nevertheless in my opinion it is difficult to evaluate the microstructural and mechanical behaviour based on the limited variant of investigated material samples. Why do the Authors propose only three variants of the studied parameters?
5)    How many material samples were produced for particular variants of scanning velocities and laser power? I don’t see the standard deviation on the charts (Figure 2b, Figure 3b, Fig 6d).
6)    Figure 3 – there is information that the value of material density was estimated based on the Archimedes method.  Why the Authors didn’t use data from CT or from optical microscopy?
Based on the CT or optical microscopy observation you can find the characteristic type of material imperfections like small or large pores, and cracks that exist depending on the applied technological parameters. Furthermore, if the Authors will increase the number of evaluated variants of 3D printing parameters they will be able to find a so-called operating window which enables avoiding the following problems: lack of fusion, balling effect, and key-hole effect that is typical in SLM processes.
7)    In Figure 6b, there is a lack of units on the chart. Please add this information.
8)    Figure 9. I will recommend replacing Figure 9. It is blurred and does not allow to state the difference in the surface quality. I think it will be better to add the side view of the sample or conduct measurements and presents the difference between assumed and real dimensions.
9)    Please add some conclusions describing the relation between material porosity and adopted 3D printing parameters. In my opinion, porosity is one of the most significant problems that exist in the SLM process and very often the microstructural and mechanical properties depend on it.

Author Response

Thank you very much for your comprehensive comments and thoughtful suggestions on our manuscript. They are very helpful for us to improve the quality of the manuscript. According to these comments and suggestions, we have carefully made modifications on the original manuscript (Manuscript ID metals-1765791). Point-by-point response to reviewers' comments is appended below.

Reviewer #3

  1. Introduction: There is no clear justification why the Sn58Bi material should be implemented in Additive Manufacturing techniques and what kind of benefits can be achieved using this material in a 3D printing process. Furthermore, can you give more details about the implementation of this material in particular branches of the industry where AM technique was used? It allows better defining the scientific background of your work and better justified its importance.

Response to the comment:

Thanks a lot for your suggestions. The SnBi alloy and other low melting point alloys have aroused great interest among researchers for many years in view of their distinct applications including solder alloys, melting model casting, tube bulging, rapid mold preparation and so on [1-4]. However, the brittleness of Sn58Bi alloy is a limitation that has to be addressed for the wide applications [7], especially for as-cast Sn58Bi alloy. In fact, two methods had been adopted to solve this problem: addition of alloying elements and severe plastic deformation. And their mechanical properties improvement mechanisms were mainly due to microstructure refinement effect. However, the addition of alloying elements may change the melting point of solder paste alloy, while severe plastic deformation methods, such as equal channel angular pressing process (ECAP) and high pressure torsion (HPT) [14-15], were not suitable for the large bulk materials. So a new approach to improve mechanical performance without changing melting point of SnBi alloy is urgently needed. According to previous investigation, even small variation in the cooling rate will lead to significant changes in microstructure and properties of SnBi alloy [16], and rapid cooling process probably leading to the fine structure of SnBi alloy. As we known, the extremely rapid solidification and cooling process were the unique characteristics of AM technology including LBPBF. Thus, LBPBF technology was firstly tried to fabricate Sn58Bi specimens in order to get fine structure and better mechanical performance in this study.

We have made corresponding revision in the revised manuscript.

Relevant references:

[1] Vianco P. ; Rejent J.,; Grant R. Development of Sn-based, low melting temperature Pb-free solder alloys. Mater. Trans. 2003, 45, 765-775.

[2] Timmel K.; Eckert S.; Gerbeth G.; et al. Experimental Modeling of the Continuous Casting Process of Steel Using Low Melting Point Metal Alloys—the LIMMCAST Program. ISIJ Int. 2010, 50, 1134-1141.

[3] Ohashi T.; Liu G. Lateral extrusion of tailor welded aluminum alloy pipes with a lost core of low temperature melting alloy. J. Ach. Mater. Manuf. Eng., 2009, 32(1).

[4] Zhang H.; Wang G.; Luo Y. and Nakaga T. Rapid hard tooling by plasma spraying for injection molding and sheet metal forming, Thin Solid Films 2001, 390, 7-12

[7] Zhou S., Shen Y.A., Tiffani U., Vasanth C.S., etc. Improve mechanical properties induced by In and In&Zn double additions to eutectic Sn58Bi alloy, J. Mater. Sci. –Mater. Electron. 2019, 30, 7423-7434.

[14] Yang L.; Zhou W.; Ma Y.; Li X.; Liang Y.; Cui W.; Wu P. Effects of Ni addition on mechanical properties of SnBi solder alloy during solid-state aging. Mater. Sci. Eng. A, 2016, 633: 367-375.

[15] Zhou S.; Yang C. H.; Lin S.K.; et al. Effects of Ti addition on the microstructure, mechanical properties and electrical resistivity of eutectic Sn58Bi alloy. Mater. Sci. Eng. A, 2019, 744: 560-569.

[16] O.V. Gusakova, V.G. Sphephelevich, L.P. Shcherbachenko, Influence of the melt cooling rate on the microstructure and texture of Sn-Bi and Sn-Pb eutectic alloys, Journal of surface Investigation, X-ray, synchrotron and Neutron Techniques, 2016, 10(1):146-152.

  1. Figure 1 – there is a lack of a chart illustrating the powder diameter distribution

Response to the comment:

Thanks a lot for your suggestions. We have made corresponding revision in the revised manuscript.

  1. The quality of the figures is very low, please put them in a higher resolution.

Response to the comment:

Thanks a lot for your suggestions. We have made corresponding revision in the revised manuscript.

  1. Line 91 there is information that three different values of the scanning velocity and laser power were applied in the studies. I understand that this approach gave a 3x3 matrix of material samples, nevertheless in my opinion it is difficult to evaluate the microstructural and mechanical behavior based on the limited variant of investigated material samples. Why do the Authors propose only three variants of the studied parameters?

Response to the comment:

Thanks a lot for your suggestions. In fact, power range of 20W-90W were also tried for specimen printing in this study. However, Sn58Bi alloy power could not be melt sufficiently and bonded into bulk material at laser power of 20W at any scanning velocity (see Fig. 2a), while the heavy dark smoke appeared at laser power of 90W which may lead to the failure of scanning mirror. So the laser powers between 30W and 80W with different scanning velocity were studied in this paper.

5) How many material samples were produced for particular variants of scanning velocities and laser power? I don’t see the standard deviation on the charts (Figure 2b, Figure 3b, Fig 6d).

Response to the comment:

Thanks a lot for your suggestions. We are sorry that there are still some statements that are not enough rigorous. There were two specimens with same parameter on each panel, and two panels of specimens were fabricated for compressive test. The Figure 2b, Figure 3b were revised in the revised manuscript. We have made corresponding revision in the revised manuscript.

6) Figure 3 – there is information that the value of material density was estimated based on the Archimedes method. Why the Authors didn’t use data from CT or from optical microscopy? Based on the CT or optical microscopy observation you can find the characteristic type of material imperfections like small or large pores, and cracks that exist depending on the applied technological parameters. Furthermore, if the Authors will increase the number of evaluated variants of 3D printing parameters they will be able to find a so-called operating window which enables avoiding the following problems: lack of fusion, balling effect, and key-hole effect that is typical in SLM processes.

Response to the comment:

Thanks a lot for your comments. The resolution of micro-CT machine used in this study was about 40μm, so the defects less than 40μm could not be detected. For optical microscopy observation, it could only show the defects on a certain cross-section. But the relative density could quantitatively compare the total defects ratio of different specimens with different parameters. The CT scanning and optical microscopy observation were also used in this study to qualitatively analysis the defects in a visual way. Just as you mentioned, the all kinds of defects are also important for quality improvement of LBPBF specimen, which will be systematically studied in the future.

7) In Figure 6d, there is a lack of units on the chart. Please add this information.

Response to the comment:

Many thanks for your comments. We have made corresponding revision in the revised manuscript.

8) Figure 9. I will recommend replacing Figure 9. It is blurred and does not allow to state the difference in the surface quality. I think it will be better to add the side view of the sample or conduct measurements and presents the difference between assumed and real dimensions.

Response to the comment:

Thanks a lot for your suggestions. The side view of specimen was conducted as shown in Figure 9b, and there was not obvious different among the side surfaces of three specimen. Measurement by vernier caliper indicated the actual height of specimen B1, C2 and D3 was 9.98mm, 9.96mm and 9.87mm, when the assume heights of specimen were 10.00mm. So high energy density specimen has the lowest dimensional accuracy in height, which may due to the convex surface and high stress make powder hard to spread on top surface. We have made corresponding revision in the revised manuscript.

Figure 9. Surface of the as-printed Sn58Bi alloy specimens in different energy densities

9) Please add some conclusions describing the relation between material porosity and adopted 3D printing parameters. In my opinion, porosity is one of the most significant problems that exist in the SLM process and very often the microstructural and mechanical properties depend on it.

Response to the comment:

Thanks a lot for your suggestions. Relative density variation with the change of printing parameters was illustrated in Figure 3. The relative densities of as-printed specimens fabricated with 30W laser power were ranged from 85.0% to 93.8%, which were much smaller than other specimens. The optical pictures of cross-section in Fig. 3a showed that a mass of voids were easily observed on the surface, and the CT pictures also illustrated a mass of defects (dark shadow) existed in the specimens, which indicated that specimens had high porosity under low laser power. In fact, the voids were caused by unfused power under low laser power (see Fig. 3a), leading to the low relative density. When the laser power was higher than 40W, the relative densities of specimens dramatically increased to the value around 98%. As one of solder paste materials, the low melting point (138°C) and excellent weldability of Sn58Bi alloy make itself easy to be shaped through LBPBF. CT pictures in Figure 3b showed that the defect bigger than 40μm was not detected within specimens of B, C and D group. The relative densities of specimens with 40-50W were even higher than relative density of as-cast alloy, and the maximum relative density (specimen C1) could reach 99.1%. On the other hand, the low vaporization temperature of Sn58Bi alloy could cause an increase in micropores when laser molten pool temperature increase, slightly decreasing the relative density of specimen with 80W. Therefore, it could be considered that laser power of 40-60W is more helpful to get higher density of as-printed alloy.

Round 2

Reviewer 1 Report

Dear authors,

the implemented changes appear reasonable and the amended manuscript appears to have reached an acceptable standard in terms of its content.

A few minor remarks:

- please double-check the abbreviation LBPBF , shouldn't PBF-LB be used instead?

- occasionally the space between the number and unit is missing, e.g. page 3, line 125: 0.06mm  or on page 5, line 171: '50W'

- please have the English grammar checked and corrected before the manuscript is published

Reviewer 2 Report

The paper is suitable for publication in its current form.

Reviewer 3 Report

Dear Authors,

I still have the impression that based on the modified version of the Introduction it is still difficult to state where a 3D printed Sn58Bi alloy can be used (types of industry branches). Nevertheless, the proposed modifications are acceptable.

The modified version of the conclusions better justified the correlation between material porosity and applied 3D printing parameters.

I don't have any objections to other sections of this manuscript. In my opinion, it can be published in the Metal journal.